# Adaptive Dendritic Cell-Negative Selection Method for Earthquake Prediction

**Wen Zhou** [1,*], **Wuyang Lan** [1], **Zhiwei Ye** [1,*], **Zhe Ming** [2], **Jingliang Chen** [1] and **Qiyi He** [1]

[1] School of Computer Science, Hubei University of Technology, Wuhan 430068, China
[2] School of Computer Science, Wuhan University, Wuhan 430072, China
* Correspondence: zw_mmwh@hbut.edu.cn (W.Z.); hgcsyzw@hbut.edu.cn (Z.Y.)

**Abstract:** Earthquake prediction (EQP) is an extremely difficult task, which has been overcome by adopting various technologies, with no further transformation so far. The negative selection algorithm (NSA) is an artificial intelligence method based on the biological immune system. It is widely used in anomaly detection due to its advantages of requiring little normal data to detect anomalies, including historical seismic-events-based EQP. However, NSA can suffer from the undesirable effect of data drift, resulting in outdated normal patterns learned from data. To tackle this problem, the data changes must be detected and processed, stimulating fast algorithmic adaptation strategies. This study proposes a dendritic cell algorithm (DCA)-based adaptive learning method for drift detection and negative selection algorithm (DC-NSA) that dynamically adapts to new input data. First, this study adopts the Gutenberg–Richter (GR) law and other earthquake distribution laws to preprocess input data. Then, the NSA is employed for EQP, and then, the dendritic cell algorithm (DCA) is employed to detect changes to trigger gradient descent strategies and update the self-set in NSA. Finally, the proposed approach is implemented to predict the earthquakes of MW > 5 in Sichuan and the surroundings during the next month. The experimental results demonstrate that our proposed DC-NSA is superior to the existing state-of-the-art EQP approaches.

**Keywords:** negative selection algorithm; dendritic cell algorithm; earthquake prediction

## 1. Introduction

EQP aims at predicting the specific location, magnitude and time of future earthquake events [1]. Through long-term observation, research and accumulation of experience by experts, it is found that there is an inherent relationship between the three elements of earthquakes (earthquake occurrence time, magnitude, and epicenter location) and characteristic indicators [2]. Prediction of changes using historical seismic characteristic indicators is a commonly used method for EQP.

With the development of artificial intelligence, researchers have tried to apply related algorithms to EQP, and achieved good prediction results. Large earthquakes is a small probability event with a long time interval, and the number of large earthquakes observed by modern observation equipment is relatively few. Therefore, the lack of data affects the accuracy of EQP using historical seismic data [1].

The NSA [3], which belongs to an artificial immune algorithm, simulates the principle of self–non-self identification in the biological immune system. The algorithm mimics the process of the negative selection of immature T lymphocytes in the organism during the maturation process. First, determine the self-set; then randomly generate detectors, by deleting the detectors that recognize the self, and keep the detectors that recognize the non-self. Its advantage is that it does not require prior knowledge, and only needs a limited amount of self-data as a training set to generate a large number of non-self detectors for detecting the non-self [3]. It is often used in binary classification problems, for it can identify self and non-self. In this study, predicting the occurrence of earthquakes of

moment magnitude (MW) 5.0 and above can be regarded as a binary classification problem in essence [4], and the NSA can be used to identify the occurrence of earthquakes (non-self) and the non-occurrence of earthquakes (self). As there is no need for non-self datasets (earthquakes of MW 5.0 and above) in the negative selection training process, the low prediction accuracy problem caused by the lack of large earthquake data in traditional machine learning methods can be solved.

In this study, the NSA of computer immunology is introduced to establish the EQP model. Using the self-non-self identification principle of negative selection, earthquakes with a predefined threshold and above magnitudes that did not occur are regarded as self, and earthquakes with a predefined threshold and above magnitudes are regarded as non-self. Since only the self data are used as the training set, the influence of the lack of large earthquake data on the training effect is reduced, and the purpose of improving the prediction accuracy of large earthquakes is achieved. Specifically, the gradient descent that encompasses DCA for data drift identification is used for the optimization of NSA. The DCA is executed in an online fashion to detect changes from the input data. A detected data drift will stimulate the optimization process. In the experimental part of this study, the MW 5.0 is used as our prediction threshold.

The main contributions of this study are described as follows:

- A NSA model with the ability of adaptive learning is proposed, and it can adapt to changes in the environment and context.
- DCA detects data drifts in the context of trigger-adaptation strategies, namely, the gradient descent is used to optimize the radius parameter of NSA.
- The proposed DC-NSA is implemented to the historical seismic events in Sichuan and the surrounding areas.

The remainder of this study is summarized as follows: Section 2 reviews the related works; the proposed DC-NSA framework is described in Section 3; the experimentation part, including baselines and comparison analysis, are demonstrated in Section 4; and Section 5 introduces the conclusion and future works.

## 2. Related Works

Various machine learning methods have been implemented in EQP using historical seismic events, including statistic methods, ANN-based approaches, deep learning, and artificial immune methods [5,6].

Several rule-based approaches have been proposed to implement EQP tasks. For instance, Dehbozorgi and Farokhi [7] first introduced neuro-fuzzy to predict short-term earthquake via historical seismic events. The time, location, seismic magnitude, depth, statistical, and entropy parameters, are adopted to predict whether an earthquake will occur in the following five minutes. The association rule mining is applied to predict a subsequent earthquake with reference to the historical seismic events [8]. A spatial analysis of magnitude distribution for EQP using ANFIS based on automatic clustering is proposed to predict earthquakes of magnitudes higher than MW 5 in Indonesia [9]. Rule-based approaches are popular due to their ease of use and flexibility.

Many studies, such as [10–12], have been proposed to deal with EQP. Of these, artificial neural networks (ANNs) are the widely used approaches due to their ability to engage in self-learning and handle complex problems. However, it suffers from the lack of training data when implemented for EQPs. Shi et al. [13] first introduced ANN in the EQP context and established the relationships between magnitude and earthquake epicentral intensity; nevertheless, the method achieves poor performance. Many research works have followed this method using various ANNs. For instance, a support vector regressor and hybrid neural network (SVR-HNN) are adopted to predict earthquakes in [14]; this approach uses maximum relevance and minimum redundancy criteria for the relevant indicators' extraction. The authors provide sixty seismic features for EQP and achieve encouraging prediction results. However, this work does not analyze seismic-related and non-seismic-related anomalies.



There are also many shallow machine learning works that focus on EQP. For example, in [15], a method to discover clustering-based patterns and predict medium-large earthquakes is proposed. This approach adopts maximum likelihood to estimate the b-value of seismic data. The authors confirm that the b-value can be regarded as an earthquake precursor, and K-means achieves good performance. However, it only considers the b-value as a seismic precursor, and thus cannot truly reflect the complexity of an earthquake and affect the prediction performance. In [16], the principal-component-analysis-based random forest (PCA-RF) was introduced to execute data dimension reduction and generate new datasets to generalize existing prediction models, and the experimental results demonstrate that the average accuracies of these approaches have been improved. Unfortunately, the differences in geological structure hinder their universality. Asim et al. [17] adopted support vector machine (SVM) and random forest to predict earthquake activities in Cyprus, while further proving that random forest is most suitable for magnitude thresholds of MW 3.0 and MW 3.5. Notably, however, random forest produces overfitting.

Several deep learning approaches have been proposed to implement EQP tasks. For instance, DeVries et al. [18] introduced a deep-learning method to determine a static stress-based criterion that can predict aftershock locations without presupposing the fault orientation. Moreover, it provides improved aftershock prediction location and identifies physical quantities that control seismic triggering when the earthquake cycle is active; however, it requires a large amount of training samples. Huang et al. [19] adopted CNN for continuous EQP using historical seismic events in Taiwan, and identified its temporal pattern, which may be useful for further EQP. However, this research conducts no analysis regarding seismic indicators. Wang et al. [20] introduced LSTM to learn the spatio-temporal relationships among earthquakes and further proved its the robustness and effectiveness. This approach can be used for EQP, even in areas without seismic sensors; however, it is computationally expensive and time-consuming.

Various artificial immune approaches have been implemented in the EQP context. For instance, in [21], NSA is adopted to reduce the impact of inadequate earthquake data on the performance in training phase. First, eight seismic indicators proposed in [11] served as the input of NSA. If an earthquake with a magnitude (MW $\geq$ 4.5) does not occur, it is regarded as "self"; otherwise, it is regarded as "non-self". A mature detector is then generated to detect anomalous test instances. While this approach can quickly detect earthquakes, the self-detectors are difficult to define, and the deletion of matching detectors in the detector generation phase will lead to low algorithmic efficiency. In [22], DCA is introduced for EQP. First, PCA is adopted to map these indicators to the safe signal (SS), pathogen-associated molecular pattern (PAMP) and danger signal (DS) of DCA; here, PAMP denotes the existence of anomalous indicators (indicating strong seismic activity), SS indicates that the possibility of normal is relatively high (that is, the seismic activity is weak), and DS shows that the possibility of an earthquake is high. DC then randomly samples antigens and signals to produce cumulative co-stimulatory molecules (csm), a semi-mature signal (semi), and a mature signal (mat). When the cumulative csm exceeds a given migration threshold, DC begins to migrate. If the cumulative semi exceeds the cumulative mat, the DC is semi-mature; otherwise, it is mature. Finally, the number of times each antigen is presented to be normal and anomalous is calculated, and the degree of abnormality of the antigen is evaluated by the calculating mature context antigen value (MCAV). If MCAV exceeds an anomalous threshold, the antigen is anomalous (an earthquake occurs); otherwise, it is normal (an earthquake does not occur). In [23], a Haskell-based deterministic DCA (EQP-hDCA) was presented to predict the magnitude of earthquakes in Sichuan and surroundings with magnitudes greater than MW 4.5 in the following month. While the DCA-based EQP obtains good performance, it is affected by a serious false alarm rate, while dealing with the application, including frequent data-type switching.

To sum up, the selection of the EQP model is very important to the prediction performance. This study proposes a novel EQP method; we select the eight indicators used in

literature [2] as eigenvectors, and use negative selection to establish an EQP model. Moreover, DCA is used for the optimization of NSA. Since the detector generation process of the NSA only needs to use the self dataset, it can be used to reduce the problem of low prediction accuracy caused by the lack of large earthquake data in methods such as neural networks. Then, the DCA detects data drifts in input data to trigger gradient descent adaptation strategies, to improve the prediction accuracy.

### 3. The Proposed DC-NSA Earthquake Prediction Approach

The DC-NSA architecture for EQP is described in Figure 1, including three parts: preprocessing, NSA prediction and DC-based adaptive learning process. Firstly, the seismic event set is described as $E = \{E_1, \ldots, E_i, \ldots E_N\}$, which is the data source of seismic indicator matrix $F$, as explained in Table 1. According to [24], *Label* $= 1$ indicates that the maximum magnitude for events occurring within the prediction horizon is larger than or equal to the threshold $M_{mean}$, and *Label* $= 0$ otherwise. The input of DC-NSA is matrix $F$. Then, the corresponding DCA model executes a signal acquisition operation to sensor changes or drifts in the contextual data, and initializes the parameters of the parameters of the gradient descent. Meanwhile, the obtained signal matrix in DCA model serves as the input of NSA to execute an EQP task. Then, with the recognition of changes or drifts in contextual data, gradient descent is implemented to optimize the radius $rs$ of NSA. The output of NSA-DCA is the prediction result; 1 indicates an earthquake of magnitude equal to or greater than $M_s$ occurs, and 0 otherwise.

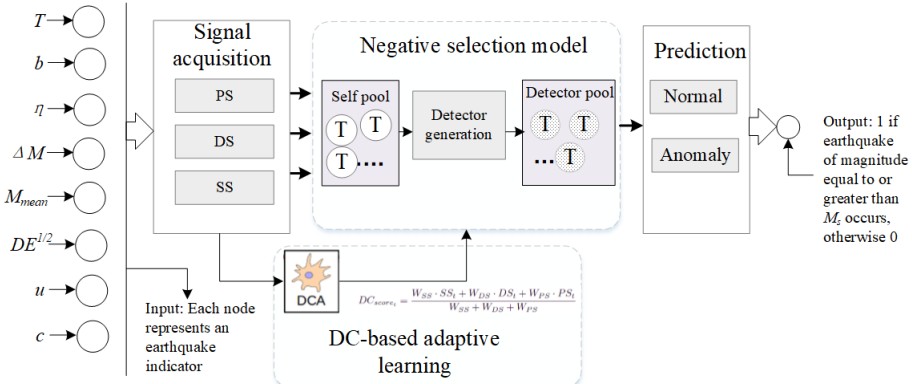

**Figure 1.** The DC-NSA architecture for EQP.

### 3.1. Seismic Indicators

The seismic indicators are calculated according to GR law, the distribution of earthquakes magnitude, and the latest EQP studies. In this study, the indicator matrix is defined as $F = \{F_1, \ldots, F_i, \ldots F_N\}, 0 < i < N$, where $N$ is the number of seismic events, while $F_i = \{b_i, \eta_i, \Delta M_i, T_{M_s i}, \mu_{M_s i}, c_{M_s i}, dE_i^{1/2}, M_{mean i}\}$ [11]. For a seismic event $i$, the indicator $b_i$ is the b-value, $\eta_i$ is the mean square deviation, $\Delta M_i$ denotes the magnitude difference between the maximum magnitude and the maximum expected magnitude observed according to GR law, $T_{M_s i}$ is the time elapsed over the last $N_i$ events with a magnitude greater than a predefined threshold ($M_s$), $\mu_{M_s i}$ is the mean time between the $N$ last $M_s$-characteristic events, $c_{M_s i}$ is the coefficient of variation of the mean time between the $N$ last $M_s$-characteristic events ($\mu_{M_s i}$), $dE_i^{1/2}$ is the rate of the square root of released seismic energy, and $M_{mean i}$ is the mean of the Richter magnitudes of the last $N$ events. Table 1 presents the formulas used to calculate these values. Here, $a_i$ denotes the cumulative frequency of earthquakes above zero, $M_i$ denotes the Richter magnitude, $n$ is the total number of earthquake events, $t_i$ denotes the observed elapsed time between characteristic events ($M_i$), and $n_c$ is the total number of characteristic events. The first three indicators and the difference $\Delta M_i$ are related to GR law, while $T_{M_s i}$, $dE_i^{1/2}$, $M_{mean i}$ are unrelated to the assumed temporal distribution of seismic magnitude; moreover, $\mu_{M_s i}$, $c_{M_s i}$ are related to the distribution of the characteristic temporal seismic magnitude.

**Table 1.** Calculation methods of each seismic indicator.

| Indicator | Calculation Methods |
|---|---|
| $b_i$ | $(n\Sigma(M_i log N_i) - \Sigma M_i(\Sigma log N_i))/((\Sigma M_i)^2 - n\Sigma M_i^2)$ |
| $\eta_i$ | $(\Sigma(log_{10} N_i - (a_i - bM_i)^2))/(n-1)$ |
| | where $a_i = \Sigma(log_{10} N_i + bM_i)/n$ |
| $\Delta M_i$ | $M_{max,observed_i} - M_{max,expected_i}$ |
| $T_{M_s\,i}$ | $t_i - t_1$ |
| $\mu_{M_s\,i}$ | $\Sigma(t_i)/n_c$ |
| $c_{M_s\,i}$ | $standard \quad deviation \quad of \quad the \quad observed times/\, u_{M_s\,i}$ |
| $dE_i^{1/2}$ | $E_i = 10^{11.8+1.5M_i} ergs$ |
| | $dE_i^{1/2} = \Sigma E_i^{1/2}/T_{M_s\,i}$ |
| $M_{mean\,i}$ | $\Sigma M_i/n$ |

### 3.2. Negative Selection Algorithm

Negative selection aims at generating tolerance to self cells, and hence, the immune system has the ability to detect unknown antigens and not to react to self-cells [3]. The input of the NSA is self set, where each self is represented by an n-dimensional vector. The detectors (antibodies) generated by the algorithm cover the non-self space. It mainly uses an n-dimensional vector to define the detector, and uses a real value to describe the detection radius of the detector. A detector can be thought of as a hypersphere. Whether the detector detects the antigen depends on the distance between the detector and the antigen (calculated using the Euclidean distance formula) and the detector radius. If the distance is less than the self-radius, which means that the two hyperspheres intersect, which means that the detector can identify self, then the detector is eliminated; otherwise, the distance between the detector and self is taken as the radius. If the mature detectors have different radii, a large range of non-self spaces can be covered by very few discriminators; meanwhile, for less non-self spaces around the self, detectors with smaller radii can be used to cover them. The mature detector generation algorithm flow is illustrated in Algorithm 1.

---

**Algorithm 1** NSA

---

Input: self $S$

Output: classification matrix

1: Initialize the number of detectors *num*, self radius *rs*, detectors set *D*, and maximum distance *distance*
2: **while** the number of detectors $< num$ **do**
3:     Generates n-dimensional vectors *d* randomly
4:     **for** each antigen *s* in training set *S* **do**
5:         Calculate the Euclidean distance *dis* between the random detector *d* and the self
6:         **if** $dis < distance$ **then**
7:             $distance = dis$
8:         **end if**
9:     **end for**
10:     **if** $rs < distance$ **then**
11:         $rd = distance - rs$
12:         Add the vector $< d, rd >$ to the mature detector set $D$
13:     **end if**
14: **end while**

---

### 3.3. Dendritic Cell Algorithm-Based NSA

In the biology immune system, dendritic cells (DCs) can identify potentially damaging foreign bodies, namely, it has strong classification abilities. Therefore, the DCA was proposed and implemented as an intrusion detection approach [25].

DCA draws on the antigen present process of DCs. DCs are able to collect and process antigens, which are molecules that stimulate immune responses, as they are in the mature state. In the immature state (iDCs), the DCs can collect diverse immune signals, including pathogen-associated molecular pattern signals (PSs), danger signals (DSs), and

safety signals (SSs). The iDC changes to a semi-mature state (smDC) or mature state (mDC) according to the concentration of immune signals received. More precisely, smDC contains more SS than PS and DS, and can be declared a security context. On the contrary, mDC exposed to a large number of PS and DS will be considered a danger environment; therefore, the antigen is eliminated. Another output is the costimulatory molecule (CSM). It executes DC migration when a predefined threshold $\gamma_{MT}$ is reached. Algorithm 2 shows the structure of DCA.

---

**Algorithm 2** DCA

---

**Input**: seismic indicators $F$.
**Output**: antigen context vectors.
**for** each antigen poll **do**
  $S_t$, $D_t$, and $P_t$ calculation by Equations (1)–(3);
**end for**
**for** each DC **do**
  antigen adaptation;
  signal fusion calculation by Equation (4) and Table 2;
  **if** $C_{CSM}$ > migration threshold **then**
    antigen adaptation;
  **end if**
  **for** the DC **do**
    **if** SEMI > $\gamma_{MT}$ **then**
      DCContext = SEMI;
    **else**
      DCContext = MAT;
    **end if**
  **end for**
**end for**
**for** each antigen **do**
  **if** MCAV > $\gamma_{FT}$ **then**
    antigen = anomalous;
    gradient descent;
  **else**
    antigen = normal;
  **end if**
**end for**

---

The anomaly metric characterizing the signal is in accordance with the distance metric, which simply calculates the difference between SS and DS in the context parameter values of time $t$ and $t-1$. The Euclidean distance between all contextual parameters at time $t$ and $t-1$ is applied for $PS$. Given a dataset of $k$ contextual features $\{B_1, \ldots, B_k\}$, where each feature variable $B_i$ can obtain a value from its own support $\chi_i$, and $n$ contextual feature instances, $b_t = (b_1, \ldots, b_k) \in \chi_1 \times \chi_2 \times \ldots \times \chi_k$, with $t \in \{1, \ldots, n\}$, $SS$, $DS$ and $PS$ signals are calculated as

$$S_t = S_t - S_{t-1} \tag{1}$$

$$D_t = D_t - D_{t-1} \tag{2}$$

$$P_t = ||b'_t - b'_{t-1}||_F \tag{3}$$

where $|| \cdot ||_F$ is the Frobenius norm, and $b'_t$ is the Z-score normalized contextual feature vectors at $a_t$. After determining when the DC becomes smDC or the lifetime duration of mDC according to the concentration of its PS, DS and SS signals, check the DC, depending on $\gamma_{MT}$. The checking process can be described as

$$DC_{score_i} = \frac{w_S * S_t + w_D * S_t + w_P * P_t}{w_S + w_D + w_P} \tag{4}$$

where $w_S$, $w_D$, and $w_P$ are the weights for the *SS*, *DS*, and *PS* signals, respectively. $S_t$, $D_t$, and $P_t$ are the signal values at time $t$ computed based on Equations (1)–(3), respectively.

When the migration threshold $\gamma_{MT}$ is reached, the DC no longer acknowledges the new instance to process, classifies the antigen and removes the DC from the population. Then it creates a new DC to replace it. A time window *TW* is adopted to account for temporality relative to the DC state. Checks the fixed segment $|t - TW|$ of a DC of size *TW*, so the resulting value can consider previous values. The $\gamma_{FT}$ is a failure threshold, and it is finally possible to estimate whether a segment of size $|t - TW|$ at time $t$ indicates a change, which in this case, corresponds to a context change, and then the gradient descent should be activated. Assuming that at least one contextual feature is set by the DC as a change at time $t$, the entire system has to undergo the data drift:

$$
r_t = \begin{cases} change, if \sum_{i=1}^{k} DC_{i|t-TW|} > 1, \\ normal, otherwise. \end{cases} \tag{5}
$$

where $r_t \in r = (r_1, \ldots, r_n)$. When $r_t = change$, the NSA model corresponds to the current system state and must adapt to the new situation. Table 2 shows examples of possible values for three DCA parameters as part of the experimental setup for real test scenarios discussed later.

**Table 2.** Signal weight values.

|      | **PS** | **DS** | **SS** |
|------|--------|--------|--------|
| CSM  | 0.4    | 0.2    | 0.4    |
| SEMI | 0      | 0      | −1     |
| MAT  | 0.4    | 0.2    | 0.4    |

## 4. Experimentation

In this section, the proposed DC-NSA will be validated and compared to various machine learning approaches: namely, DCA [22], NSA [21], PCA-RF [16], back propagation neural network (BPNN) [11], RNN [2], probabilistic neural network (PNN) [12], EQP-hDCA [23], LSTM [19] and SVR-HNN [14].

### 4.1. The Dataset and Performance Measurement

Information about earthquakes from Gansu (DS1), Qinghai (DS2), Sichuan (DS3) and Yunnan (DS4) from 1 January 1990 to 13 October 2018 was obtained from the China National Earthquake Science Data Center [26]. As it is of great significance to predict earthquakes with a magnitude greater than a threshold, we selected 12,040 data points with magnitudes greater than MW 3.0 as our data source. Table 3 presents the distributions of each extracted dataset. The mean is the average of the magnitudes, and SD is the standard deviation of the magnitudes of each dataset.

**Table 3.** Analyzed seismic areas of China.

| Areas          | Number of Data | Mean | SD   |
|----------------|----------------|------|------|
| Gansu (DS1)    | 1132           | 3.51 | 0.50 |
| Qinghai (DS2)  | 2086           | 3.67 | 0.61 |
| Sichuan (DS3)  | 5433           | 3.52 | 0.49 |
| Yunnan (DS4)   | 3389           | 3.47 | 0.49 |

Table 4 lists the calculated eight seismicity indicators of Sichuan province according to Table 1, the $N$ is set to 100 [2], and the $M_s$ is set as MW 5.0.

**Table 4.** Partial eight indicators and label of Sichuan province after preprocess.

| Date | b | $\eta$ | $\Delta M$ | T | $\mu$ | c | $dE^{1/2}$ ($\times 10^{10} ergs$) | $M_{mean}$ | C |
|---|---|---|---|---|---|---|---|---|---|
| 201810 | 0.77 | 0.001 | $-0.21$ | 267 | 20 | 1.44 | 0.015 | 3.27 | 0 |
| 201607 | 0.81 | 0.004 | 0.66 | 167 | 14 | 0.67 | 0.032 | 3.151 | 1 |
| ... | ... | ... | ... | ... | ... | ... | ... | ... | ... |
| 201109 | 1.02 | 0.005 | $-0.21$ | 87 | 10 | 0.61 | 0.033 | 3.29 | 0 |
| 200102 | 0.75 | 0.003 | $-0.01$ | 21 | 0 | 0 | 3.555 | 3.007 | 0 |

Through the seismic indicators calculation method described in Section 3, all indicator matrices *F* of DS1, DS2, DS3 and DS4 are calculated, and the box plot is drawn as shown in Figure 2. It can be seen from the figure that there are outliers in the original seismic data and the calculated seismic indicators.

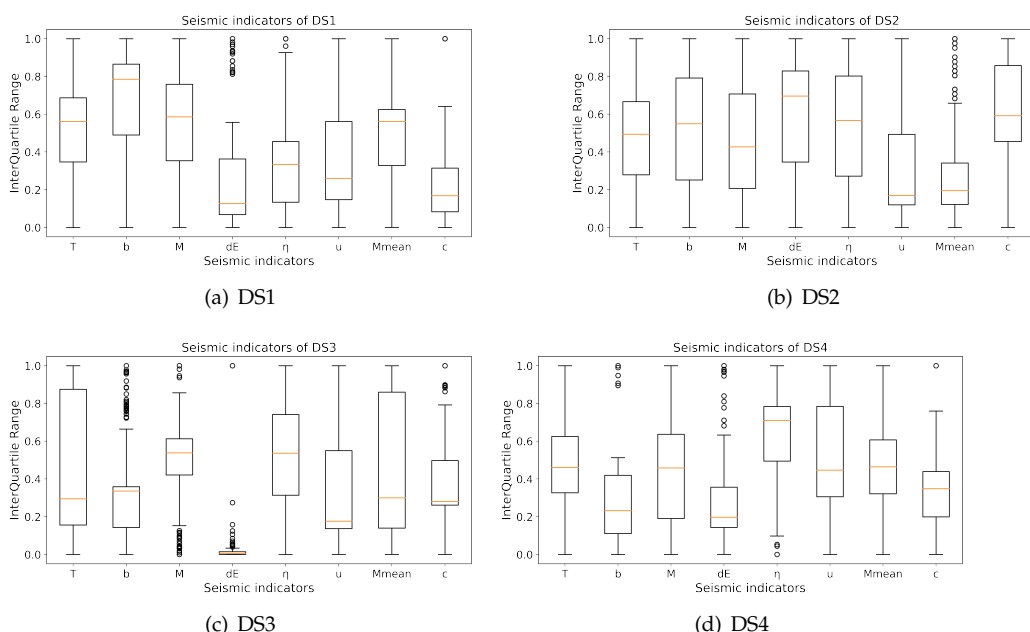

**Figure 2.** Box plot of the eight seismic indicators distribution of each dataset.

### 4.2. The Prototype Implementation

In our experiments, the input signals are normalized between 0 and 1; therefore, the DC weight matrix is set according to Table 2; the $M_s$ is MW 5.0. Other parameters are as follows: $\gamma_{MT} = 0.5$, $\gamma_{FT} = 0.4$, the learning rate, epochs, and batch size are set to 0.002, 1, and 10, respectively. DCA and EQP-hDCA use PCA as the signal mapping method, create 100 DCs for the pool, and randomly select 10 DCs to sample each antigen. NSA uses the Euclidean distance to measure the similarity between antigen and antibody, and the self radius is set to 0.02. For PCA-RF, the parameters are set as maxDepth = 0, numFeatures = 0, numTrees = 10. The BPNN in our experiment has three layers: input layer, hidden layer and output layer. The learning rate indicator is set to 0.0003, and the number of training periods is set to 100 periods. The weights in the multi-layer perceptron were learned using standard gradient descent. The RNN uses sigmoid as the activation function. The batch size is set to 128, the learning rate indicator is set to 0.001, and the number of training time periods is set to 100 time periods. The weights in the multi-layer perceptron are set for learning using the randomly optimized A method (Adam). The PNN in our experiment has an embedded layer, a product layer and three hidden layers. In LSTM, the learning rate is set to 0.001, and momentum is used to accelerate model convergence, and is set to 0.9. We use 0.0005 weight attenuation, and the abscission rate of the two abscission layers is 0.5. Other parameter settings of the comparison methods can be found in the relevant literature.



### 4.3. Verification Indicators

The EQP according to historical seismic events is a classification task. Therefore, the following statistical indicators are used to assess the results of the proposed and comparison algorithms: false acceptance rate (FAR), Matthews correlation coefficient (MCC), R-score (R), accuracy (ACC), predictive positive value (PPV), negative predictive value (NPV), recall rate (Rn), specificity (S), and area under the curve (AUC). Apparently, the averages (Avgs) of PPV, NPV, Rn and S are calculated to provide a overall prediction quality measure [11].

### 4.4. Results Analysis and Comparison

The experimental results of DS1, DS2, DS3 and DS4 are demonstrated in this section. The results of each domain are analyzed separately. Tables 5–8 show the results of DS1 to DS4, respectively. In all tables, the number in bold indicates the best result among all the methods.

There are only 1132 instances in DS1 area. With reference to DS1 (see Table 5), when using performance evaluation indicators to evaluate all algorithms, the performance is not very good. AUC is only a little higher than 0.5 (random classification results), and R is also poor. However, in such a case, we found that many indicators of DC-NSA are better than NSA. At the same time, in most evaluation indicators, the results are not as good as those of DCA, and are similar to those of DCA in R. Therefore, the results of DC-NSA are not competitive on this dataset.

The domain DS2 has a higher SD, and Table 6 summarizes the results of DS2. It is worth noting that compared with other methods, the predicted FAR of DC-NSA is limited, while the value of R is 1.00, which is a very ideal feature of EQP method. Therefore, the higher SD may achieve higher performance.

There are 5433 instances in DS1 area. The specific results of DS3 are shown in Table 7. The difference among Rn, MCC and R between DC-NSA and other algorithms is very obvious, with the difference exceeding 0.18 units. The difference between the second-best classifier of Rn (EQP hDCA) and the second-best classifier of MCC (LSTM) exceeds 0.28 units. Compared with the second-best classifier of R (EQP hDCA), the difference exceeds 0.27 units. Therefore, the higher SD may achieve higher performance.

Table 8 shows the results of DS4. DC-NSA provides better results on most evaluation indicators, with the smallest FAR and a R value of 0.96. When the other nine algorithms predict DS4, the overall difference between DC-NSA and EQP hDCA is not particularly obvious. In the aspect of index R (the key index of EQP algorithm), NSA is obviously weaker than other algorithms.

From a joint analysis of the above-mentioned tables, since DC-NSA achieves the best results on most indicators, such as R and Avg, we can conclude that DC-NSA is the most suitable classifier for these datasets. Moreover, it can be found that the datasets with higher SD or lager number of instances may achieve higher performance.

**Table 5.** Comparison between DC-NSA and other approaches when predicting DS1. The best results are in bold.

|  | PPV | NPV | Rn | S | FAR | MCC | AUC | Avg | R |
|---|---|---|---|---|---|---|---|---|---|
| DC-NSA | 0.43 | 0.73 | 0.71 | 0.45 | 0.23 | 0.14 | 0.59 | **0.59** | **0.50** |
| DCA [22] | 0.43 | 0.75 | **0.72** | 0.53 | 0.25 | 0.18 | 0.60 | 0.61 | 0.47 |
| NSA [21] | 0.21 | 0.52 | 0.21 | 0.51 | 0.38 | 0.02 | 0.51 | 0.36 | −0.17 |
| PCA-RF [16] | 0.55 | 0.61 | 0.60 | 0.21 | **0.42** | **0.23** | 0.52 | 0.49 | 0.18 |
| BPNN [11] | **0.64** | 0.63 | 0.35 | 0.15 | 0.37 | **0.23** | **0.61** | 0.44 | −0.02 |
| RNN [2] | 0.27 | 0.75 | 0.68 | 0.32 | 0.24 | 0.04 | 0.48 | 0.51 | 0.44 |
| PNN [12] | 0.32 | 0.64 | 0.13 | **0.63** | 0.33 | −0.02 | 0.51 | 0.48 | −0.20 |
| EQP-hDCA [23] | 0.32 | 0.74 | 0.70 | 0.37 | 0.26 | 0.06 | 0.53 | 0.53 | 0.44 |
| LSTM [19] | 0.42 | **0.81** | 0.69 | 0.31 | 0.23 | 0.12 | 0.54 | 0.56 | 0.46 |
| SVR-HNN [14] | 0.38 | 0.76 | 0.67 | 0.29 | 0.28 | 0.09 | 0.51 | 0.53 | 0.39 |

**Table 6.** Comparison between DC-NSA and other approaches when predicting DS2. The best results are in bold.

|  | PPV | NPV | Rn | S | FAR | MCC | AUC | Avg | R |
|---|---|---|---|---|---|---|---|---|---|
| DC-NSA | 0.84 | **1.00** | **1.00** | **0.96** | **0.00** | **0.91** | **0.97** | **0.95** | **1.00** |
| DCA [22] | 0.17 | 0.82 | 0.50 | 0.52 | 0.18 | −0.01 | 0.46 | 0.50 | 0.32 |
| NSA [21] | 0.00 | 0.51 | 0.49 | 0.00 | 0.42 | 0.00 | 0.45 | 0.25 | −0.42 |
| PCA-RF [16] | 0.61 | 0.56 | 0.58 | 0.77 | 0.41 | 0.02 | 0.43 | 0.63 | 0.37 |
| BPNN [11] | 0.67 | 0.96 | 0.67 | 0.04 | 0.04 | 0.63 | 0.82 | 0.58 | 0.63 |
| RNN [2] | **0.86** | 0.98 | 0.90 | 0.91 | 0.01 | 0.87 | 0.90 | 0.91 | 0.89 |
| PNN [12] | 0.84 | 0.92 | 0.81 | 0.82 | 0.06 | 0.81 | 0.86 | 0.85 | 0.75 |
| EQP-hDCA [23] | 0.73 | **1.00** | **1.00** | 0.93 | **0.00** | 0.83 | **0.97** | 0.92 | 1.00 |
| LSTM [19] | 0.78 | 0.82 | 0.91 | 0.87 | 0.06 | 0.82 | 0.85 | 0.85 | 0.85 |
| SVR-HNN [14] | 0.69 | 0.80 | 0.86 | 0.79 | 0.12 | 0.68 | 0.66 | 0.79 | 0.74 |

**Table 7.** Comparison between DC-NSA and other approaches when predicting DS3. The best results are in bold.

|  | PPV | NPV | Rn | S | FAR | MCC | AUC | Avg | R |
|---|---|---|---|---|---|---|---|---|---|
| DC-NSA | **0.91** | **1.00** | **1.00** | **0.91** | **0.00** | **0.92** | **0.96** | **0.93** | **1.00** |
| DCA [22] | 0.33 | 0.78 | 0.24 | 0.15 | 0.22 | 0.10 | 0.62 | 0.37 | 0.02 |
| NSA [21] | 0.33 | 0.69 | 0.73 | 0.49 | 0.28 | 0.18 | 0.62 | 0.56 | 0.45 |
| PCA-RF [16] | 0.32 | 0.73 | 0.43 | 0.21 | 0.32 | 0.12 | 0.44 | 0.42 | 0.11 |
| BPNN [11] | 0.56 | 0.90 | 0.50 | 0.08 | 0.10 | 0.44 | 0.71 | 0.51 | 0.40 |
| RNN [2] | 0.45 | 0.66 | 0.66 | 0.55 | 0.33 | 0.02 | 0.52 | 0.58 | 0.14 |
| PNN [12] | 0.47 | 0.79 | 0.64 | 0.59 | 0.19 | 0.38 | 0.51 | 0.62 | 0.45 |
| EQP-hDCA [23] | 0.46 | 0.91 | 0.82 | 0.63 | 0.09 | 0.41 | 0.73 | 0.71 | 0.73 |
| LSTM [19] | 0.65 | 0.79 | 0.81 | 0.53 | 0.11 | 0.65 | 0.71 | 0.70 | 0.70 |
| SVR-HNN [14] | 0.59 | 0.62 | 0.73 | 0.49 | 0.19 | 0.59 | 0.63 | 0.61 | 0.54 |

**Table 8.** Comparison between DC-NSA and other approaches when predicting DS4. The best results are in bold.

|  | PPV | NPV | Rn | S | FAR | MCC | AUC | Avg | R |
|---|---|---|---|---|---|---|---|---|---|
| DC-NSA | 0.77 | 0.94 | 0.95 | **0.85** | **0.01** | **0.79** | **0.90** | **0.85** | **0.96** |
| DCA [22] | 0.33 | 0.58 | 0.50 | 0.59 | 0.42 | −0.09 | 0.43 | 0.50 | 0.08 |
| NSA [21] | 0.00 | 0.32 | 0.32 | 0.00 | 0.55 | 0.00 | 0.23 | 0.16 | −0.55 |
| PCA-RF [16] | 0.31 | 0.59 | 0.53 | 0.23 | 0.43 | 0.02 | 0.50 | 0.41 | 0.10 |
| BPNN [11] | **1.00** | 0.10 | 0.15 | 0.00 | 0.90 | 0.12 | 0.59 | 0.31 | −0.75 |
| RNN [2] | 0.63 | 0.67 | 0.59 | 0.60 | 0.28 | 0.55 | 0.57 | 0.62 | 0.31 |
| PNN [12] | 0.66 | 0.68 | 0.53 | 0.61 | 0.29 | 0.46 | 0.55 | 0.62 | 0.24 |
| EQP-hDCA [23] | 0.67 | **0.98** | **0.97** | 0.77 | 0.02 | 0.70 | 0.87 | **0.85** | 0.95 |
| LSTM [19] | 0.35 | 0.31 | 0.48 | 0.42 | 0.21 | 0.57 | 0.46 | 0.39 | 0.37 |
| SVR-HNN [14] | 0.37 | 0.56 | 0.61 | 0.59 | 0.16 | 0.65 | 0.56 | 0.53 | 0.45 |

## 5. Conclusions and Future Work

This paper described an EQP method by integrating DCA and NSA. The main contribution of this paper is that it presents a novel DC-NSA approach to make a more suitable EQP model. Firstly, DC-NSA preprocesses earthquake indicators using the GR law and other earthquake magnitude distribution techniques, and obtains a seismic indicator $F$. Further, the corresponding DCA model executes a signal acquisition operation to sensor changes or drifts in the contextual data, and initializes the parameters of the parameters of the gradient descent. Meanwhile, the obtained signal matrix in DCA model serves as the input of NSA to execute an EQP task. Then, with the recognition of changes or drifts in contextual data, gradient descent is implemented to optimize the radius $rs$ of NSA. Finally, this study used the historical seismic events in Sichuan and surroundings as our experimental data, and compared the proposed approach with DCA, NSA, PCA-RF, BPNN, RNN, PNN, EQP-hDCA, LSTM, and SVR-HNN, using the PPV, NPV, Rn, S, FAR, MCC, AUC, Avg, and R as comparison criteria. The experimental results demonstrate that our proposed DC-NSA is superior to the compared approaches.

In this study, the proposed method is derived from the function of T cell and DCs in the immune system; unlike traditional EQP methods, which lacks adaptively, the proposed approach implements the change adjustments based on the input sample. Meanwhile, the DC-NSA can handle the problem that NSA suffers from the normal patterns learned from data that have become obsolete.

However, this study only focuses on the historical seismic events, neglecting the observed data, which are also important for EQP. Therefore, we will adopt the observed data with much more instances than our experimental data. Moreover, future work will focus on enriching NSA and applying it to other earthquake-prone areas. We will further test NSA to improve the AIS detection performance through subsequent experiments.

**Author Contributions:** Validation, W.L.; Investigation, Z.M. and J.C.; Writing—original draft, W.Z.; Writing—review & editing, Z.Y. and Q.H.. All authors have read and agreed to the published version of the manuscript.

**Funding:** This research was funded by NSFC grant number 61877045 and 62202147, Key projects of Scientific Research Project of Hubei Provincial Department of Education grant number D20191406.

**Data Availability Statement:** Not applicable.

**Acknowledgments:** The authors want to thank NSFC- http://www.nsfc.gov.cn/ (accessed on 10 November 2022) for the support through Grants Number 61877045 and 62202147, Key projects of Scientific Research Project of Hubei Provincial Department of Education for their support through Grant Number D20191406.

**Conflicts of Interest:** The authors declare no conflict of interest.

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
