# Peer review of "Adaptive Dendritic Cell-Negative Selection Method for Earthquake Prediction"

_electronics, doi:10.3390/electronics12010009_

Round 1
Reviewer 1 Report
This research work applies a negative selection algorithm that encompasses a dendritic cell algorithm for earthquake prediction in four seismic areas in China.
There are several unsupported issues arising in the manuscript both in terms of the methodology used as well as in terms of the underlying geoscience.
- Firstly, there is no discussion whether the data are characteristic of the system. Are the four areas under ivestigation distinct seismic regions? Are they part of the same underground fault network? Relying on few data for earthquake prediction is not very wise.
- Threshold magnitude has been sellected at M? 5.0. Is that ML or Ms or... and why M5.0?
- Same applies for the prediction horizon also arbitrarily set at 1 month. In large earthquakes foreshocks and/or aftershocks of the main earthquake can occure a month in advance or a month later depending on the underlying geodesy.
- The manuscript combines together two existing methodologies and aplies very few data to it and yields some results. In my opinion this is still preliminary work.
- In terms of the data, 1132 to 5433 earthquakes depending on the region under investigation is extremely low and definately not characteristic of the system. You need to identify pottential distinct seismic regions in the vicinity/vicinities under investigation and multiply your data set.
- The system is poorly described and again there no justification on the selection of parameters such as learning rates, epochs, batch size and so on. Setting up a systems to get some results again is preliminary work.
- There is an outline of over a page of results with no discusion, only a gathering of "titles" in a quick paragraph.
- Conclusions are practically missing all together.
For all the above reasons i must recomend that the manuscript is rejected.
Author Response
Dear Reviewer,
On behalf of my co-authors, we thank you very much for your positive and constructive comments and suggestions on our manuscript with ID electronics-2056749.
We have studied the reviewers' comments carefully and have made revisions in the manuscript according to your comments. Please see the attachment.
Again, we would like to express our great appreciation to you for comments on our manuscript.
Thank you and best regards,
Wen Zhou

Reviewer 2 Report
So far, many methods have been tried and applied for earthquake predictions. However, very important steps have not been passed. This article is a study conducted with an artificial intelligence method based on the biological immune system using negative selection algorithm (NSA) from a different perspective. I think it will inspire new studies on this subject.
Author Response
Dear Reviewer,
On behalf of my co-authors, we thank you very much for your positive and constructive comments and suggestions on our manuscript with ID electronics-2056749.
Thank you and best regards,
Wen Zhou
Reviewer 3 Report
The paper proposes an adaptive learning method for the detection of drifts by using the so-called dendritic cells. An associated negative selection algorithm is proposed for dynamically adapting the new input data. Globally, it is an interesting work, that investigates an updated problem. The techniques used are well-controlled and already accompanied by easy algorithms. The conclusion, the abstract, and the introduction are acceptable, reminiscent of some minor improvements. I think that the abstract is somehow long, contrary to the conclusion which is somehow short. We prefer the converse to hold. The literature review and the comparison with existing works may also be more developed.
Author Response
Dear Reviewer,
On behalf of my co-authors, we thank you very much for your positive and constructive comments and suggestions on our manuscript with ID electronics-2056749.
We have studied the reviewers' comments carefully and have made revisions in the manuscript according to your comments. Please see the attachment.
Again, we would like to express our great appreciation to you for comments on our manuscript. We look forward to hearing from you soon.
Thank you and best regards,
Wen Zhou

Round 2
Reviewer 1 Report
-